# Efficacy Assessment of Biosynthesized Copper Oxide Nanoparticles (CuO-NPs) on Stored Grain Insects and Their Impacts on Morphological and Physiological Traits of Wheat (*Triticum aestivum* L.) Plant

**DOI:** 10.3390/biology10030233

**Published:** 2021-03-17

**Authors:** Ali A. Badawy, Nilly A. H. Abdelfattah, Salem S. Salem, Mohamed F. Awad, Amr Fouda

**Affiliations:** 1Botany and Microbiology Department, Faculty of Science, Al-Azhar University, Nasr City, Cairo 11884, Egypt; ali.abdalhalim@azhar.edu.eg (A.A.B.); salemsalahsalem@azhar.edu.eg (S.S.S.); 2Agriculture Research Centre, Plant Protection Research Institute, Dokki, Giza 12619, Egypt; dr.nilly@yahoo.com; 3Department of Biology, College of Science, Taif University, P.O. Box 11099, Taif 21944, Saudi Arabia; m.fadl@tu.edu.sa

**Keywords:** biosynthesis, CuO-NPs, grain insects, *Triticum aestivum* L., antioxidant enzymes

## Abstract

**Simple Summary:**

In the current study, copper oxide nanoparticles (CuO-NPs) were successfully formed through the reduction of CuSO_4_·5H_2_O by the activity of *Aspergillus niger* strain (G3-1) metabolites. The as-formed CuO-NPs were characterized by UV–visible (UV–vis) spectroscopy, X-ray diffraction (XRD), transmission electron microscopy (TEM), scanning electron microscopy–energy dispersive spectroscopy (SEM-EDX), Fourier-transform infra-red (FT-IR) spectroscopy, and X-ray photoelectron spectroscopy (XPS). Data showed the efficacy of fungal metabolites to fabricate crystallographic, spherical CuO-NPs with sizes of 14.0 to 47.4 nm at varied bending energies. The biosynthesized CuO-NPs exhibited activity against two wheat grain insects, *Sitophilus granarius* and *Rhyzopertha dominica*, in a size- and dose-dependent manner. Moreover, the lower CuO-NP concentrations had beneficial effects on the morphological and physiological aspects of wheat plants, without displaying any toxicity or repression of growth.

**Abstract:**

Herein, CuO-NPs were fabricated by harnessing metabolites of *Aspergillus niger* strain (G3-1) and characterized using UV–vis spectroscopy, XRD, TEM, SEM-EDX, FT-IR, and XPS. Spherical, crystallographic CuO-NPs were synthesized in sizes ranging from 14.0 to 47.4 nm, as indicated by TEM and XRD. EDX and XPS confirmed the presence of Cu and O with weight percentages of 62.96% and 22.93%, respectively, at varied bending energies. FT-IR spectra identified functional groups of metabolites that could act as reducing, capping, and stabilizing agents to the CuO-NPs. The insecticidal activity of CuO-NPs against wheat grain insects *Sitophilus granarius* and *Rhyzopertha dominica* was dose- and time-dependent. The mortality percentages due to NP treatment were 55–94.4% (*S. granarius*) and 70–90% (*R. dominica*). A botanical experiment was done in a randomized block design. Low CuO-NP concentration (50 ppm) caused significant increases in growth characteristics (shoot and root length, fresh and dry weight of shoot and root, and leaves number), photosynthetic pigments (total chlorophylls and carotenoids), and antioxidant enzymes of wheat plants. There was no significant change in carbohydrate or protein content. The use of CuO-NPs is a promising tool to control grain insects and enhance wheat growth performance.

## 1. Introduction

Nanotechnology is an interdisciplinary science, growing rapidly due to its incorporation into various scientific sectors, such as medicine, engineering, pharmacy, industry, wastewater treatment, paper preservation, and agriculture [1,2,3]. Nanoparticles (NPs) are aggregates of atoms or molecules at a nano-size, ranging from 1 to 100 nm [4]. Nanomaterials have unique physical and chemical properties that differ from their bulk forms [5]. These properties include a larger surface to volume ratio in comparison with bulk materials, size, shape, crystal structure, thermal stability, charge, and zeta potential [6,7]. NPs are synthesized using various approaches, including chemical, physical, and biological methods. Biological synthesis is preferred over chemical and physical methods [8]. The harnessing of metabolites generated from different biological entities, including prokaryotic and eukaryotic organisms (e.g., bacteria, actinomycetes, fungi, yeast, and plants), to fabricate metals and their oxide NPs has recently begun to generate more interest [8]. This may be attributed to the advantages that these methods offer, such as biocompatibility, safe and easy handling, large-scale production potential, rapid development, the avoidance of hazard by-products, and being environmentally friendly.

Copper oxide nanoparticles (CuO-NPs) are highly interactive nanomaterials that display various biological activities, such as antibacterial and antifungal activity, antioxidant properties, drug delivery, and cytotoxic efficacy against tumor and cancer cells [9,10]. CuO-NPs have been fabricated either extra- or intracellularly through different biological entities, such as bacterial, fungal, actinomycetes, algae, and plants. The harnessing of fungal species as reducing, capping, and stabilizing agents to fabricate NPs is most interesting, because of various secreted metabolites, high metal accumulation, and scalability [11].

Wheat (*Triticum aestivum* L.) crop belongs to the family Graminae (Poaceae) and is an annual, large shrub plant commonly used as food. It is the most valuable food and strategic cereal crop in Egypt and ranks third globally, after corn and rice, as the most productive grain [12]. Wheat can grow under different environmental conditions, and this allows for extensive cultivation as well as long-term food storage. In general, around 70% of wheat crops are used for human nutrition, 19% for animal feed, and the remaining 11% for industrial applications, including in biofuels [13].

Pest infestation of wheat crops can start in the field and continue through storage until the grains are processed for consumption. Stored grain pests cause high economic losses by feeding on stored grains, and they harm public health by contaminating food. *Sitophylus granarius* and *Rhyzopertha dominica* are major stored wheat pests worldwide that cause the most quantitative and qualitative losses [14]. Immature insects can develop inside the grain. The larvae and adults can feed on grain kernels and leave behind dust; thin brown shells or a musty odor are often associated with the infestations of these insects [15]. Since residues from synthetic insecticides are a potential hazard for mammals, and insect resistance to synthetic insecticides is an ongoing challenge [16], there is a need to search for a new, readily available, affordable substance that can act as an insecticide while being less toxic to mammals and less detrimental to the environment.

Copper (Cu) is a vital element when combined with specific proteins and enzymes that have essential functions in the growth and nutrition of plants [17]. It is involved in cell wall metabolism, mitochondrial respiration, photosynthetic electron transport, oxidative stress responses, and hormone signaling [18]. Copper can also cause enhanced production of bioactive compounds, reactive oxygen species (ROS) plant growth inhibition, and altered root systems [19,20,21].

This study aimed to synthesize CuO-NPs using an eco-friendly approach by using metabolites present in biomass filtrate of *Aspergillus niger*. Characterizations were performed with UV–vis spectroscopy, X-ray diffraction (XRD), transmission electron microscopy (TEM), SEM-EDX, FT-IR, and X-ray photoelectron spectroscopy (XPS). The efficacy of green-synthesized CuO-NPs as insecticides against wheat grain insects *Sitophylus granarius* and *Rhyzopertha dominica* was evaluated. The treatment of wheat plants (*Triticum aestivum* L.) with biosynthesized CuO-NPs and their resultant growth performance was also investigated.

## 2. Materials and Methods

### 2.1. The Fungal Strain Used for Biosynthesis of CuO-NPs

*Aspergillus niger* strain G3-1 was isolated from desert soil samples from El-Wahat, Giza, Egypt (GPS N: 2°21′38″ E: 28°55′56.3″) and cultivated on potato dextrose (PD) agar media. The fungal strain was subjected to cultural, morphological, and molecular identification based on internal transcribed spacer (ITS) sequencing analysis, as previously described by Fouda et al. [22]. The gene sequence obtained from molecular identification was placed in the gene bank with accession number ky465752.

### 2.2. Biosynthesis of CuO-NPs

A three-disk (0.7 mm in diameter) sample of old culture *A. niger* strain G3-1 was inoculated onto 100 mL of PD broth media and incubated for 6 days at 28 °C ± 2 °C and under 150-rpm shaking conditions. At the end of the incubation period, the cultivated medium was subjected to centrifugation to separate fungal biomass, which was washed three times with distilled water to remove any medial components. Next, the washed fungal biomass (15 g) was mixed with 100 mL distilled water for 48 h, then centrifuged to collect biomass filtrate (i.e., supernatant without any fungal biomass), which was used later to fabricate CuO-NPs as follows: 2 mM of the metal precursor CuSO_4_·5H_2_O was mixed with 100 mL biomass filtrate for 24 h at 30 °C ± 2 °C and under 150-rpm shaking conditions. The resultant greenish dark NPs were collected and oven-dried at 200 °C for 24 h.

### 2.3. Characterization of Fungal-Mediated CuO-NPs

#### 2.3.1. UV–Vis Spectroscopy Analysis

The resultant greenish color, which formed due to CuO-NP biosynthesis, was monitored by UV–vis spectroscopic analysis (Jenway 6305 Spectrophotometer) at wavelength 200–800 nm, to detect surface plasmon resonance (SPR).

#### 2.3.2. X-ray Diffraction (XRD) Analysis

The crystallinity of the fungal-mediated CuO-NPs was analyzed using XRD patterns with the X-ray diffractometer X’Pert Pro (Philips, Eindhoven, Netherlands). The 2*θ* was in the range of 4 °C–70 °C. Ni-filtered Cu Ka radiation was used as the X-ray source, while the voltage and current were 40 kV and 30 mA, respectively [23].

#### 2.3.3. Transmission Electron Microscopy (TEM) Analysis

The formed size and specific shape of the fungal-mediated CuO-NPs were examined using TEM (JEOL 1010 Japan) with an accelerating voltage of 200 KV, magnification mode of 600× to 500,000×, and tilt angles of X-tilt +/−60°. For this purpose, a drop of NP solution was loaded on a carbon-coated copper grid. The excess NP solution was removed by placing the grid on blotting paper. Afterward, the grid was left to dry before being placed on a specimen holder [24,25].

#### 2.3.4. Energy-Dispersive Spectroscopic Analysis (SEM-EDX)

The quantitative elemental composition of the fungal-mediated CuO-NPs was explored using Quanta 250FEG (Thermo Fisher Scientific, Waltham, MA. USA) (accelerating voltage: 20.00 kV; magnification mode: 3000×; with detector BSED). The Quanta FEG is integrated with an energy dispersive spectroscopy (EDX) detector (FEI company subsidiary of Thermo Fischer Scientific, Inc., Backscattered electron (BSE) detector; high electron beam at 2.5 nm at 30 KV) [26].

#### 2.3.5. Fourier-Transform Infrared (FT-IR) Spectroscopy

FT-IR analysis (Agilent system Cary630 FT-IR model) was used for the biomass filtrate of the fungal strain and the biosynthesized CuO-NPs, to detect the various functional groups accountable for reducing, capping, and stabilizing the NPs. The transmittance mode spectra were measured over a range of 4000–400 cm^−1^, with 12 scans.

#### 2.3.6. X-ray Photoelectron Spectroscopy (XPS) Analysis

XPS spectra were analyzed using an ESCALAB 250XI^+^ instrument (Thermo Fischer Scientific, Inc., Waltham, MA, USA) connected with a monochromatic X-ray with Al Kα radiation (1486.6 eV). The analysis was conducted under the following conditions: spot size was set to 500 µm, samples were prepared under pressure adjusted to 10^−8^ mbar, the energy was calibrated with the Ag3d_5/2_ signal (∆BE: 0.45 eV) and the C 1s signal (∆BE: 0.82 eV), and the full- and narrow-spectrum pass energy were 50 eV and 20 eV, respectively [27].

### 2.4. Insecticidal Bioassay

The insects *Sitophilus granarius* (adult wheat weevil) and *Rhyzopertha dominica* (small grains borer) were reared for several generations without exposure to any insecticide on their food (wheat grains). The experiment and insect rear were performed in the laboratory of the Department of Stored Products and Grains Pests, Plant Protection Research Institute, Agriculture Research Centre, Dokki, Giza, Egypt. One hundred insects were placed in a 500-mL jar containing 200 g wheat grains. After one week, the adults were removed, and the jar was left unchanged until the next generation emerged. To start the experiment, the jar was covered with gauze and tightly closed with a rubber band. The wheat grains were sterilized at 55 °C for 6 h, to eliminate any hidden infestation. Five different concentrations (100, 150, 200, 250, and 300 mg) of CuO-NPs were mixed with 100 g of wheat grains. Twenty adults of each species were placed separately in each replicate, and three replicates were used for each treatment. Mortality rates were calculated after 2.0, 4.0, 6.0, 8.0, and 10.0 days; then, all treatments were left until the first (f1) progeny emerged.

The reduction percentages in the progeny offspring were calculated using the following equation [28]:(1)$$\begin{array}{c} \;\;\;\;\mathrm{Reduction}\;\mathrm{percentages}\;(\%)\;\arcsin\;\theta \\ =\frac{\mathrm{No}.\mathrm{of}\;\mathrm{progeny}\;\mathrm{of}\;\mathrm{contol}-\mathrm{No}.\mathrm{of}\;\mathrm{progeny}\;\mathrm{of}\;\mathrm{treated}}{\mathrm{No}.\mathrm{of}\;\mathrm{progeny}\;\mathrm{of}\;\mathrm{control}} \end{array}$$

### 2.5. Effect of Biosynthesized CuO-NPs on Wheat Plant Growth Performance

#### 2.5.1. Field Experiment Design

A field experiment was carried out during the winter of 2019/2020 to evaluate the impact of biologically-synthesized CuO-NPs on the growth and metabolism of wheat (*Triticum aestivum* L.) plants. The experiment took place in the botanical garden of the Botany and Microbiology Department, Faculty of Science, Al-Azhar University, Cairo, Egypt. Wheat grains (cv. Masr 1) were obtained from the Agriculture Research Center, Ministry of Agriculture, Giza, Egypt. Uniform grains were cultivated in 5 ridges. The first ridge represented the control plants. The second-ridge plants were foliar-sprayed with 50 ppm CuO-NPs, while the third-ridge plants were foliar-sprayed with 100 ppm CuO-NPs. The fourth ridge contained plants whose seeds were soaked in 50 ppm CuO-NPs before sowing. The fifth ridge had plants whose seeds were soaked in 100 ppm CuO-NPs before sowing. The plants were irrigated whenever required. The physicochemical characteristics of the soil used in the field experiments were classified as sandy soil, containing sand (94.26%), silt (4.35%), and clay (1.39%). The plant samples were collected at the vegetative stage after 6 weeks of sowing for analysis.

#### 2.5.2. Morphological Measurements

Five plants were randomly collected from each treatment to estimate root length, shoot length, number of leaves, root fresh weight, root dry weight, shoot fresh weight, and shoot dry weight.

#### 2.5.3. Physiological Measurements

Leaf pigments (chlorophylls and carotenoids) in the fresh leaves were assessed according to [29]. Briefly, one gram of fresh leaves was extracted using 100 mL of 80% acetone (*v*/*v*). The homogenate was filtered, and then the filtrate was diluted to a total volume of 100 mL using 80% acetone. The absorbance of the extract was measured in three replicates at 665, 649, and 470 nm using a spectrophotometer (Jenway 6305 Spectrophotometer).

The contents of the total soluble carbohydrates were estimated using the anthrone technique according to [30]. Briefly, 1 g of dried plant tissue was extracted with 5 mL of 2% phenol and 10 mL 30% trichloroacetic acid. The mixture was shaken and kept overnight before being filtered. Each sample filtrate was filtered with charcoal. The clear filtrate was diluted quantitatively with distilled water. Next, 2 mL of diluted filtrate was mixed with 4 mL of freshly prepared anthrone reagent (2 g anthrone/L of 95% pure sulfuric acid). The developed color was measured using a spectrophotometer at 620 nm. The experiment was performed in triplicate.

The method described by [31] was used to assay the contents of soluble proteins. Briefly, 1 g of the dried sample was extracted in 5 mL of 2% phenol and 10 mL distilled water. The mixture was shaken and kept overnight before being filtered. Each sample was filtered using charcoal. The reagents used were solution A (2% Na_2_CO_3_ in 0.1 N NaOH), solution B (0.5 g CuSO_4_ in 1% sodium potassium tartrate), solution C (50 mL of solution A were mixed with 1 mL of solution B), and solution D (diluting Folin reagent with distilled water in the proportion of 1:3). In a test tube, 1 mL of the clear filtrate of the plant sample was added to 5 mL of solution (C). The contents of the tube were mixed and left to stand for 10 min. Next, 0.5 mL of solution (D) was rapidly added and mixed with the tube contents, then left to stand for an additional 30 min. The optical density of the resultant color was read at the wavelength of 750 nm. The experiment was performed in triplicate.

Antioxidant enzyme superoxide dismutase (EC1.15.1.1) activity was determined by measuring the repression of pyrogallol auto-oxidation as described in [32]. The solution (10 mL) consisted of 3.6 mL of distilled water, 0.1 mL of enzyme, 5.5 mL of 50 mM phosphate buffer (pH 7.8), and 0.8 mL of 3 mM pyrogallol (dissolved in 10 mM HCl). The rate of pyrogallol reduction was measured at 325 nm with a UV spectrophotometer.

To estimate the activity of peroxidase (EC1.11.1.7), the method outlined by [33] was used as follows: 5.8 mL of 50 mM phosphate buffer (pH 7.0) was mixed with 0.2 mL of the enzyme extract and 2 mL of 20 mM H_2_O_2_, followed by adding 2 mL of 20 mM pyrogallol. The rate of increase in absorbance of pyrogallol was determined using a UV–vis spectrophotometer within 60 s at 470 nm.

The method provided by [34] was used to estimate the activity of ascorbate peroxidase (EC1.11.1.11) as follows: 0.5 mM ASA, 0.8 mL of potassium phosphate buffer (50 mM, pH 7), 0.1 mM H_2_O_2_, and 0.2 mL enzyme extract were mixed. The changes in absorbance were read at 290 nm.

### 2.6. Statistical Analysis

Data were statistically analyzed using SPSS v17 (SPSS Inc., Chicago, IL, USA). One-way analysis of variance (ANOVA) was used to investigate the efficacy of CuO-NPs on wheat growth performance. A posteriori multiple comparisons were done using Tukey’s range tests at *p* < 0.05. All results are the means of three to six independent replicates, as specified above.

## 3. Result and Discussion.

### 3.1. Aspergillus Niger-Mediated Biosynthesis of CuO-NPs

In this study, the metabolites of fungal isolate *Aspergillus niger* G3-1 were a catalyst to the formation of CuO-NPs. Once the fungal biomass was mixed with CuSO_4_·5H_2_O, its color was changed from colorless to greenish-brown. These color changes were due to the reduction of Cu^+2^ to nanoscale CuO via enzymes and proteins in the fungal biomass [35,36]. The bioreduction processes were dependent on the number of metabolites present in the biomass filtrate, which enhance the production process, decrease the aggregation, and produce a smaller size [37,38].

### 3.2. Characterization of CuO-NPs

#### 3.2.1. UV–Vis Spectroscopy Analysis

*A. niger*-mediated biosynthesis of CuO-NPs was confirmed by UV–vis spectroscopic analysis, which showed two peaks. The first peak was related to cuprous oxide (Cu_2_O) at 245 nm, and the second was for cupric oxide (CuO) at 360 nm (Figure 1A). Data in this study are compatible with Rabiee et al. [39], who reported that the UV–vis spectra of CuO-NPs synthesized by *Achillea millefolium* leaf extract also had two peaks, with one at 250 nm for cuprous oxide and one at 365 nm for cupric oxide. In contrast, Yin et al. [40] showed that the UV–vis spectrum of CuO nanocrystals has three absorption peaks at 260 nm, 340 nm, and 630 nm. He reported that the observed peaks at 260 nm and 340 nm were due to the band-to-band transition of Cu_2_O nanocrystalline, while the peak at 630 nm was attributed to the bandgap transition of CuO. Moreover, the UV–vis spectrum of CuO-NPs fabricated by aqueous extract of brown algae *Bifurcaria bifurcata* showed two absorption bands at 260 nm and 650 nm, corresponding to the formation of Cu_2_O and CuO, respectively [41]. The results of the current study indicate the successful formation of Cu_2_O and CuO in the colloidal NP solution, which conforms with the various published studies [39,42], and this finding was confirmed by XPS analysis. These characteristic patterns of as-formed CuO-NPs were due to the efficacy of water-soluble metabolites present in the biomass filtrate to reduce copper ions [41]. The broadest peaks in UV–vis spectra could be due to the broad size distribution of CuO-NPs [39]. The color change was correlated with the surface plasmon resonance (SPR) of formed NPs, which confirms the efficacy of metabolites involved in *A. niger* biomass to reduce metal Cu^+2^ to the nanoscale [43,44].

#### 3.2.2. XRD Analysis

XRD patterns are crucial tools for the confirmation of the crystalline nature of green-synthesized CuO-NPs. As seen in Figure 1B, XRD-based CuO-NP characterization exhibited eight peaks at 2θ values: 32.9°, 35.8°, 39.2°, 58.6°, 61.9°, 66.5°, 68.4°, and 72.8°. These were assigned to planes (110), (−111), (111), (−202), (202), (113), (311), and (221), respectively. The visualized XRD peaks were matched with JCPDS number 01-080-0076 for crystallographic CuO-NPs [36]. Gopinath et al. [45] reported the successful fabrication of the crystalline, monoclinic phase of CuO-NPs at the same XRD planes using *Tribulus terrestris* aqueous fruit extract, which is consistent with our study. The average size of crystalline CuO particles can be calculated using the Debye–Scherrer equation [46]:(2)D=0.94λ/βcosθ
where D is the average particle size; 0.94 is the Scherrer’s constant; λ is the X-ray wavelength; β and θ are the full widths at half maximum XRD line and half diffraction angle, respectively. In this study, the average CuO-NP size obtained from XRD analysis ranged between 3.0 and 14.1 nm. The presence of other peaks in the XRD pattern was due to impurities that could be attributed to the precipitation of metabolites in the biomass filtrate that acted as capping agents [47].

#### 3.2.3. TEM, SEM-EDX Analyses

The biosynthesized CuO-NPs were characterized with TEM and SEM-EDX analysis to investigate their shape and size and to perform quantitative elemental analysis of NPs. Figure 2 shows the successful fabrication of spherical CuO-NPs by harnessing metabolites involved in the biomass filtrate of *A. niger* G3-1, with the average size ranging between 14.0 and 47.37 nm. According to XRD and TEM analysis, the size of biosynthesized CuO-NPs did not exceed 50 nm, which is considered a small size. This phenomenon could be useful for biotechnological and biomedical applications in which NP size needs to be consistently small. As NP size decreases, their potential application increases [26,43,48]. The obtained results are compatible with Manjari et al. [49], who successfully synthesized spherical CuO-NPs of 36–54 nm using the flower extract of *Aglaia elaegnoidea*. Recently, the metabolites secreted by fungal strain *Aspergillus terreus* AF-1 were utilized as a reducing and capping agent to synthesize spherical CuO-NPs with a size range of 11–47 nm [50].

In this study, the CuO-NP size obtained by TEM analysis was larger than that obtained by XRD. Each particle contains an amorphous and a crystal domain, and while TEM analysis results in the sum of these two domains as particle size, XRD only gives results for the crystal domain. Moreover, XRD measures the core of coated NPs, but not the surface coating, while TEM analysis measures overall NP size [51].

The qualitative and quantitative analyses of biosynthesized CuO-NPs were achieved using SEM-EDX to detect elemental compositions of the sample with relative percentages, such as weight and atomic % (Figure 2B,C). Notably, the highest EDX peaks reflected element concentrations. The EDX spectra contained mainly the elements Cu and O, with weight percentages of 62.96% and 22.93%, respectively. The presence of S may have been due to the conjugation of CuO-NPs with fungal biomolecules found in the biomass filtrate, such as amino acids (cysteine or methionine) and phospholipids [52]. In line with this, Hassan et al. [47] reported that the main EDX spectra peaks for CuO-NPs synthesized by endophytic actinomycetes were Cu and O, with the presence of other minor peaks related to biomolecules in the actinobacterial filtrate, which conjugated with CuO-NPs.

#### 3.2.4. FT-IR Analysis

The functional groups responsible for reducing and stabilizing CuO-NPs, as well as those found in NPs, were characterized using FT-IR analysis. As seen in Figure 3, fungal biomass filtrate had varied peaks at 659, 1630, 2302, and 3331 cm^−1^. The broad absorption peaks at 3331 and 1630 cm^−1^ corresponded to N-H vibration bending amine and stretching C=C conjugated alkene, respectively [47]. A strong broad peak at 659 cm^−1^ referred to the C=C bending of alkene, while the peak at 2302 cm^−1^ corresponded to the stretching of the S-H thiol group in proteins [53]; meanwhile, IR of the biogenic CuO-NPs showed different peaks at 3595 cm^−1^ (O-H stretching, alcohol), 3378 cm^−1^, and 3305 cm^−1^ (N-H stretching, aliphatic primary amines, and O-H broad stretching alcohol, respectively) [54,55]. Other IR CuO-NPs peaks appeared at 2925 cm^−1^ (C-H stretching, medium, alkane), 2030 cm^−1^ (C=C=C, stretching, medium, allene), 1635 cm^−1^ (C=C, medium, alkene), 1087 cm^−1^ (strong C-O stretching, secondary alcohol), 1035 cm^−1^ (medium C-N, amine), and 871 cm^−1^ (strong C-H bending) [56,57]. Interestingly, the IR peaks at 477 and 596 cm^−1^ corresponded to the formation of metal oxide (Cu-O) [47,58]. The presence of functional groups in the biomass filtrate displayed by the FT-IR spectra could be responsible for reducing, capping, and stabilizing the biogenic CuO-NPs.

#### 3.2.5. X-ray Photoelectron Spectroscopy (XPS)

XPS survey of the green-synthesized CuO-NPs is given in Figure 4. The presence of copper is confirmed by Cu 3*p3*, Cu *3s*, Cu LM1, Cu LM2, Cu LM3, Cu 2p1, and Cu 2p2. Besides Cu, other peaks for different elements related to biomolecules in the biomass filtrate were investigated, including O (2s, 1s KL1, and KL2), carbon (as C1s and CKL1), Na (as Na 2s and Na KL1), and S (as S2p and S 2s). C1 was split into four individual peaks, which were identified as C(H, C) at 284.25 eV; C(S, NH, NH_2_) [59] at 285.25 eV; C(-O, =N) [60] at 286.15 eV; and C(-O-C, ≡N) [61] at 287.7 eV. O1s was deconvoluted into five peaks at 531.23 eV for O(Cu(II), C, H) [62,63,64]; 532.6 eV for O-(N); 535.75 eV for C-O-C; 529.25 eV for O-Cu(I); and 534.1 eV for Na KL1, which indicated the presence of Cu as +1 and +2 oxides. There were three individual peaks for N 1s at 399.42 eV, 398.5 eV, and 401.15 eV for N(C, =C, H), N≡C, and N_tert_, respectively [65,66,67], which indicated that it was present as amine (i.e., 1°, 2°, and 3° amines). Cu2p was deconvoluted into six peaks at 934.58 eV and 936.9 eV for Cu(II) 2p3/2; 932.55 eV for Cu(I) 2p3/2; 937.95 eV, 938.6 eV, 939.15 eV, 940.7 eV, and 943.35 eV for Cu (II) Satellite; 950.2 eV, 951.25 eV, and 952.2 eV for Cu(I)2p1/2; 954.45 eV for Cu(II)2p1/2; and 959.4 eV, and 962.35 eV for Cu (II) satellite, which indicated the presence of Cu as (I and II) oxides.

### 3.3. The Assessment of Green-Synthesized CuO-NPs as an Insecticide

Insects are one of the main causes of damage to stored grains. In the past, control of these insects has been accomplished using various chemical compounds, which have negative impacts on grain and users in the long term [68]. Natural plant extractions, such as essential oils, can be used as alternatives to chemical insecticides [69]. Green insecticides are ecofriendly, safe, biocompatible, and easy to apply [70], particularly green insecticides made of metal and metal oxide NPs. In this study, the potentiality of as-formed CuO-NPs as an insecticide was investigated against two wheat grain insects.

Our data analysis showed the potential that CuO-NPs have as an insecticidal agent against *S. granarius* at different concentrations (Table 1). As CuO-NP concentration and exposure time increased, the mortality rate was also increased. The mortality percentages ranged from 55% to 94.4% at CuO-NP concentrations from 100 to 300 mg/100 g of wheat grains after 10 days. At these concentrations, there was a 65–80% reduction in the F1 generation.

Data represented in Table 2 display the effect of CuO-NPs as an insecticidal agent against *R. dominica* at different concentrations. Data analysis revealed that the mortality rates were in the range of 70% to 90% at NP concentrations 100 to 300 mg/100 g wheat after exposure time 10 days. At these concentrations, there was a 33–100% reduction in the F1 generation, which ranged from 33% to 100%.

It can be concluded from our results that biosynthesized CuO-NPs can be used as insecticides against stored grain insects, as the mortality of adults of both species was observed at all concentrations. These results coincide with El-Saadony et al. [71], who concluded that biosynthesized Cu-NPs were toxic against the stored grain pest *Tribolium castaneum*, with an LC50 value of 37 ppm after 5 days of treatment. A negligible effect was observed with the chemical synthesis of Cu-NPs at the same concentration. Malaikozhundan et al. [72] stated that the biopesticide effect of *Bt*-ZnO-NPs was tested against the pulse beetle (*Callosobruchus maculatus*). Treatment with *Bt*-ZnO-NPs reduced the fecundity (eggs laid) and hatchability of *C. maculatus* in a dose-dependent manner. Thus, *Bt*-ZnO-NPs were highly effective in the control of *C. maculatus* since they caused 100% mortality at 25 μg·mL^−1^.

It can also be concluded that the biological production of NPs could be considered an ecofriendly approach due to its avoidance of hazardous chemicals and harsh conditions used during chemical and physical fabrications. This conclusion is supported by Selvan et al. [73] and Gunalan et al. [74], who demonstrated that both biogenic and green synthesis of Cu-NPs was ecofriendly and inexpensive. Green manufacturing of CuO-NPs is a safe, cost-effective, and ecofriendly process against the most critical economic insects, such as *Tribolium castaneum*, which causes significant losses in crops that enter human and animal food chains [71]. The mechanism of mortality may be the entry of the NPs into the insect’s cuticle, which blocks skin pores’ respiratory openings and leads to suffocation and death. The toxic effect may also be the result of the particles entering the insect’s blood, which leads to poisoning and the loss of appetite. This phenomenon is consistent with Yamanaka et al. [75], who stated that toxicity may be attributed to the ability of NPs to pass through epithelial and endothelial cells via transcytosis. Furthermore, NPs can proceed easily along the dendrites, axons, blood, and lymphatic vessels, causing oxidative stress and other effects [76].

### 3.4. The Efficacy of Green-Synthesized CuO-NPs on Wheat Growth Behaviors

#### 3.4.1. Morphological Measurements

The impact of soaking and foliar spray with biologically synthesized CuO-NPs on the morphological attributes of wheat plants was illustrated in Table 3. The application of 50 ppm and 100 ppm CuO-NPs (via foliar or soaking method) increased the growth indices (lengths, fresh weights and dry weights of roots and shoots, and the number of leaves) of wheat plants compared to the control. Data analysis showed that the root length of wheat plants treated with 50 and 100 ppm CuO-NPs via the foliar spray method (13.16 ± 0.57 and 12.94 ± 0.58 cm, respectively) was higher than plants treated with soaking methods at the same concentrations (10.6 ± 0.48 and 9.82 ± 0.63 cm). In the same regard, the foliar spray method was better than the soaking method in root fresh and dry weights and in shoot dry weight. In general, lower concentrations of CuO-NPs were markedly better than higher concentrations for both methods. The toxicity of high CuO-NP concentrations in plants has been recorded previously by [77,78]. Moreover, Rajput et al. [79] reported that the germination of *Hordeum sativum* L., as well as the shoot and root length, were decreased with increased doses of Cu bulk and CuO-NPs. Furthermore, they found that the high concentrations of Cu bulk and CuO-NPs have destructive effects on the cellular structure of *Hordeum sativum* L. and concluded that the CuO-NPs are better than the Cu bulk structure and a low concentration is preferred over a high concentration for plant treatment [79]. This finding increases the strength of our study of the use of low CuO-NP concentrations. The toxicity of high concentrations could be attributed to the liberation of Cu ions from CuO-NPs in aqueous solutions, which is a toxicity factor in plants [80]. Several studies have reported an increase in plant growth criteria in response to the application of CuO-NPs at small concentrations [81,82,83,84]. Additionally, Yasmeen et al. [85] have reported that the growth of wheat (*Triticum aestivum*) plants were increased when plants were soaked with Cu-NPs. Zuverza-Mena et al. [86] found that fresh and dry weights of cilantro (*Coriandrum sativum*) plants were increased when the plants were grown in soil amended with CuO-NPs at 80 mg/kg. Hafeez et al. [87] observed that soils amended with copper NPs at concentrations from 10 to 30 ppm significantly increased the growth of wheat. In another study, the root and shoot length of the eggplant seedlings was increased as compared to control plants due to treatment with a lower CuO-NP concentration (100 mg/L) [88]. The increased length of the seedlings at these concentrations indicates the growth-promoting ability of CuO-NPs at lower concentrations [88]. These positive results were probably due to copper’s role as an essential micronutrient for plant growth.

#### 3.4.2. Leaf Pigments

Leaf pigments were also observed (Figure 5A,B). The results showed significant increases in the total chlorophylls and carotenoids in response to all treatments of CuO-NPs. At 50 ppm, the highest significant increases in total chlorophylls (23% and 10.5%) and carotenoids (20.2% and 12.3%) were observed. In a recent study by [83], lower concentrations of CuO-NPs showed increased leaf pigments (chlorophylls and carotenoids) in tomato plants (*Lycioersicum esculentum*). Moreover, lower concentrations of CuO-NPs stimulated photosynthesis up to 14%, as compared with the control level of tomato plants [82]. This increase in photosynthesis rate might be due to the increased biological and chemical activities of metals at the nanoscale and the correlative impact of nutrients (magnesium, iron, zinc, sulfur, etc.) on plants. Young [89] indicated that the promotion of carotenoids plays a significant role in scavenging ROS and protects the plant from the predicted stress. Metal NPs can also enhance the structure of chlorophyll and facilitate the formation of pigments and metabolic activities [90,91,92]. It has been noted that copper is a microelement needed in small quantities for normal plant growth, due to it being a structural element in regulatory proteins, an activator of several enzymatic reactions, and a participant in the photosynthetic electron transport chain [93].

#### 3.4.3. Carbohydrate and Protein Contents

The carbohydrate and protein contents of wheat plants were also assessed (Figure 6A,B). Non-significant increases were found in response to the foliar or soaking treatments of CuO-NPs at 50 and 100 ppm. The foliar treatment at 50 ppm recorded the highest increase in carbohydrate (21%) and protein (13%) contents as compared to the control. Carbohydrate and protein contents were reduced with increased CuO-NP concentrations (100 ppm) but did not decline below the control level. It is well noted that Cu ions in small concentrations can stimulate plant growth due to its role as an essential crop micronutrient [94,95], its direct participation in redox enzymes, and its involvement in chlorophyll synthesis and protein and carbohydrate metabolism [96]. Our findings are in line with the investigation of Singh et al. [82] on tomato plants, who found that a lower concentration of CuO-NPs (10 mg·L^−1^) exhibited an increase (7%) in carbohydrate content. Additionally, Ochoa et al. [97] observed non-significant increases in the protein contents of green pea (*Pisum sativum*) plants after adding 50 and 100 mg/kg soil of CuO-NPs. Wheat plants subjected to CuO-NPs in concentrations ranging from 0.2 ppm to 1.0 ppm exhibited significant increases in different agronomic attributes [98].

#### 3.4.4. Antioxidant Enzyme Activities

The activities of antioxidant enzymes were influenced in wheat plants exposed to 50 or 100 ppm of CuO-NPs for both application methods (Figure 7A–C). All tested concentrations of CuO-NPs caused improvements in the levels of superoxide dismutase (SOD), peroxidase (POD), and ascorbate peroxidase (APX). The most significant enhancements were recorded for the foliar treatment with CuO-NPs at 50 ppm in SOD activity (approximately 42.9%) and the foliar treatment with CuO-NPs at 100 ppm in POD and APX activities (approximately 42.4% and 41.9%, respectively). Our results are in harmony with several investigations that showed the promotional role of CuO-NPs in antioxidant enzymes [82,83,99]. In some previous studies, APX and POD activities were found to be increased because of CuO-NP application [100,101]. Recent results from Sarkar et al. [102] demonstrated that APX and SOD activities were increased in lentil (*Lens culinaris*) plants treated with different concentrations of CuO-NPs (0.01, 0.025, and 0.05 mg·mL^−1^). Copper, an important micronutrient, participates in several physiological and biochemical processes, is required as an activator for superoxide dismutase [103], and is integrated into various enzymes, and it thus has a critical role in plant growth and nutrition [98]. The increased activities of antioxidant enzymes displayed a protective role against oxidative damage in plants.

According to obtained data, we can conclude that 50 ppm of CuO-NPs exhibits more positive effects on the morphological and physiological traits of the wheat plant than 100 ppm. Foliar spraying appears to be more effective than soaking, which can be attributed to the spraying method ensuring a more uniform spread over the crop canopy, in addition to eliciting a more immediate response from the crops [104]. Foliar spray showed significant superiority compared with soaking and coating methods. Sarkar et al. [105] reported that the foliar application with micronutrients could be more effective than other methods because it overcomes deficiency problems in the soil amendment or grain soaking.

## 4. Conclusions

In our investigation, we successfully biosynthesized CuO-NPs using the biomass filtrate from the *A. niger* strain (G3-1) and characterized the CuO-NPs using UV–vis spectroscopy, XRD, TEM, SEM-EDX, FT-IR, and XPS analyses. Crystallographic, spherical CuO-NPs in sizes ranging from 14.0 to 47.37 nm were synthesized at different bending energies. We found that the biosynthesized CuO-NPs exhibited high efficacy against *Sitophilus granarius* and *Rhyzopertha dominica* insects, which cause damage to stored wheat grains. Moreover, the foliar or soaking treatment of wheat plants by CuO-NPs at different concentrations did not exhibit toxicity or inhibit the growth characteristics of the wheat plants but rather displayed some enhancements. The most significant plant enhancements were recorded at lower concentrations of CuO-NPs. Morphological attributes (lengths, fresh weight, and dry weight of root and shoot, as well as leaves number) and leaf pigments (chlorophylls and carotenoids) were increased in response to CuO-NP treatments. Soluble carbohydrates and proteins were insignificantly boosted. Biosynthesized CuO-NPs were found to markedly promote and induce the activities of antioxidant enzymes. Our study recommends the use of lower concentrations of CuO-NPs in agricultural fields as insecticides that do not inhibit plant growth. Therefore, we can conclude that the positive impact of green-synthesized CuO-NPs is a promising strategy for use against insects and to enhance plant growth.

## Figures and Tables

**Figure 1 biology-10-00233-f001:**
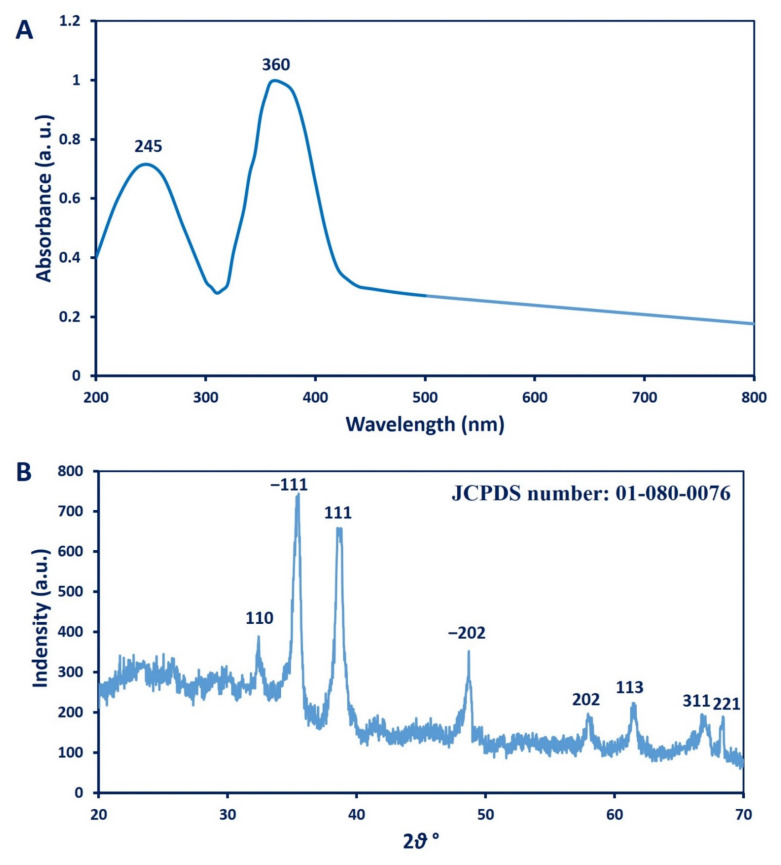
(**A**) UV–vis spectroscopic analysis of CuO-NPs synthesized by *A. niger* G3-1; (**B**) X-ray diffraction pattern of CuO-NPs showed intense peaks at specific 2 theta values.

**Figure 2 biology-10-00233-f002:**
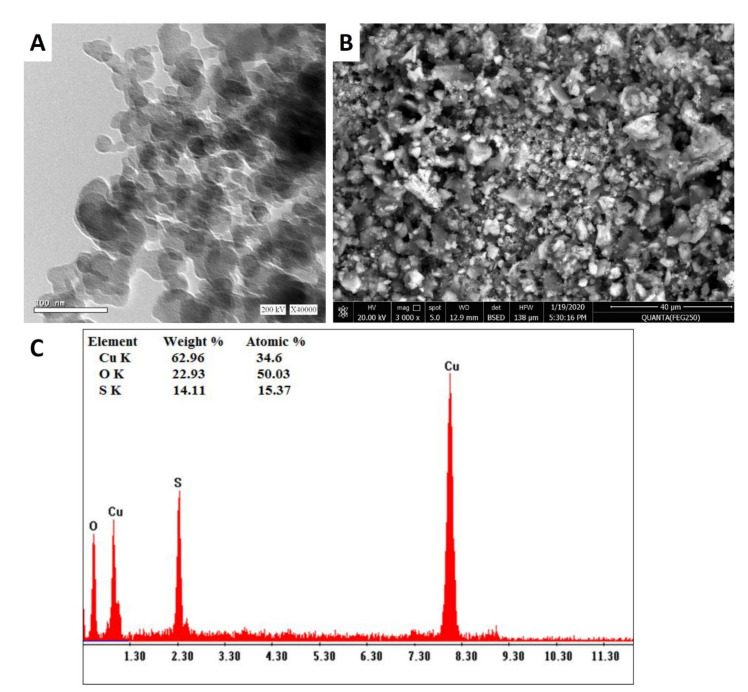
Characterization of CuO-NPs synthesized by *A. niger* G3-1; (**A**) transmission electron microscopy (TEM); (**B**) and (**C**) denote SEM-EDX spectra of biosynthesized CuO-NPs.

**Figure 3 biology-10-00233-f003:**
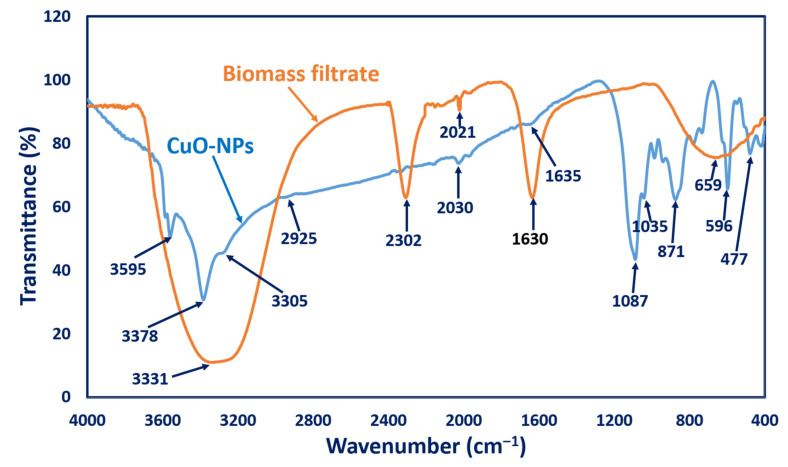
FT-IR of fungal biomass filtrate and CuO-NPs synthesized by *Aspergillus niger* G3-1.

**Figure 4 biology-10-00233-f004:**
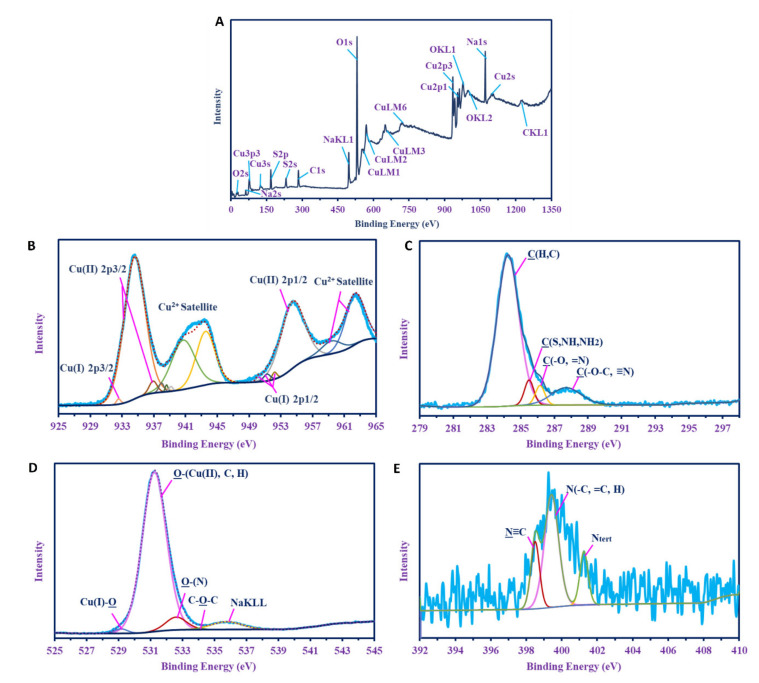
XPS analysis of CuO-NPs synthesized by *A. niger* strain G3-1. (**A**) Overall survey; (**B**) core CuO-NPs; (**C**–**E**) C 1s, O 1s, N 1s, respectively, of CuO-NPs.

**Figure 5 biology-10-00233-f005:**
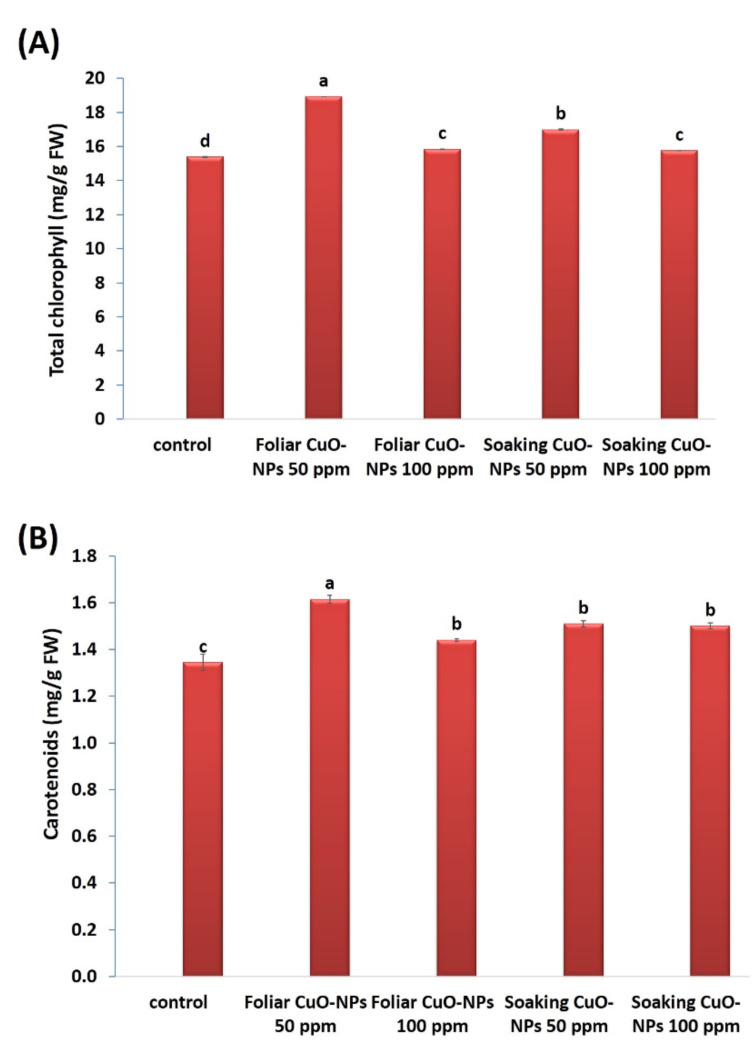
Assessing the effect of CuO-NPs on the total chlorophyll (**A**) and carotenoid (**B**) contents (mg/g FW) in wheat plants. Data are statistically different at *p* ≤0.05 by Tukey’s test, (*n* = 3); different letters on bars indicate that mean values are significantly different at the significant level of (*p* ≤ 0.05), error bars are means ± standard error (SE).

**Figure 6 biology-10-00233-f006:**
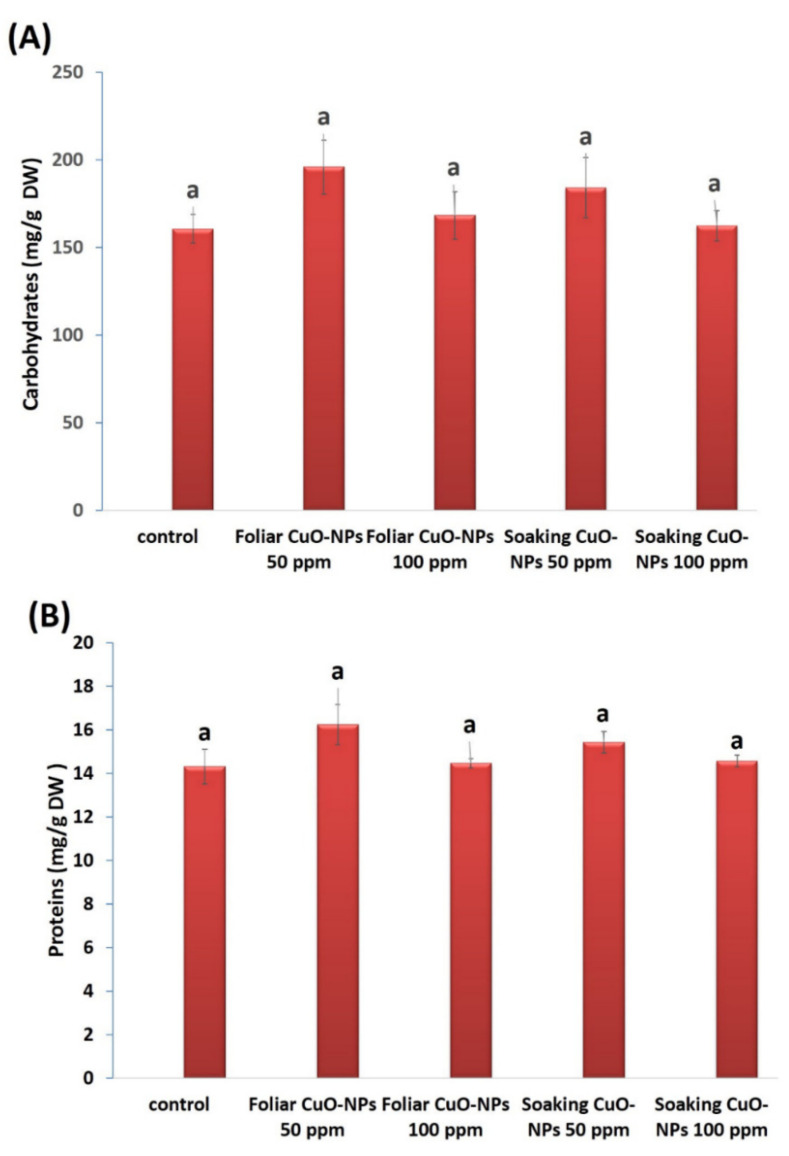
The efficacy of CuO-NPs on soluble carbohydrate (**A**) and soluble protein (**B**) contents (mg/g DW) in wheat plants. Data are statistically different at *p* ≤ 0.05 by Tukey’s test, (*n* = 3); different letters on bars indicate that mean values are significantly different at the significant level of (*p* ≤ 0.05), error bars are means ± standard error (SE).

**Figure 7 biology-10-00233-f007:**
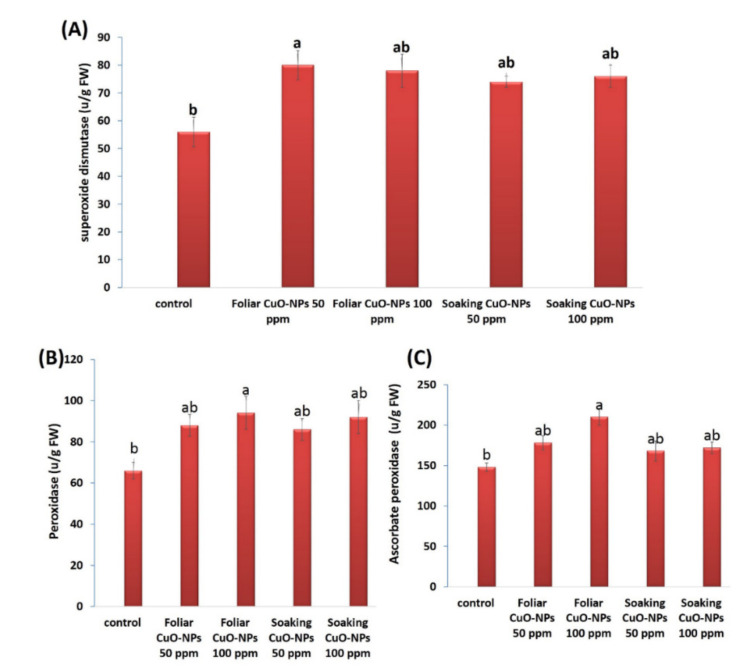
Antioxidant enzymatic activities of a wheat plant treated with CuO-NPs at two concentrations (50 and 100 ppm) by foliar spray and soaking method. (**A**) Superoxide dismutase, (**B**) peroxidase, and (**C**) ascorbate peroxidase activities (unit/g FW). Data are statistically different at *p* ≤ 0.05 by Tukey’s test, (*n* = 3); different letters on bars indicate that mean values are significantly different at a significant level of (*p* ≤ 0.05), error bars are means ± standard error (SE).

**Table 1 biology-10-00233-t001:** The efficacy of CuO-NPs synthesized using *A. niger* G3-1 against *S. granaries* insect.

NP Conc. (mg)/100 g Wheat Grains	Accumulated Mortality Percentages (%)					Reduction in F1%
	2 Days	4 Days	6 Days	8 Days	10 Days	
300	3.3 ± 1.6	36.7 ± 3.3	75.9 ± 6.7	90.8 ± 1.6	94.4 ± 2.9	80
250	2.5 ± 1.4	35 ± 2.9	72.2 ± 5.8	86.1 ± 1.4	88.9 ± 0	67.5
200	0 ± 0	30 ± 5.8	63.9 ± 4.3	66.7 ± 5.8	77.8 ± 0	65
150	0 ± 0	30 ± 7.6	63 ± 10	70.3 ± 10	72.2 ± 8.7	65
100	0 ± 0	25 ± 0	44.4 ± 0	55.6 ± 0	55.6 ± 0	70

**Table 2 biology-10-00233-t002:** The efficacy of CuO-NPs synthesized using *A. niger* G3-1 against *R. dominica* insect.

NP Conc. (mg)/100 g Wheat Grains	Accumulated Mortality Percentages (%)					Reduction in F1%
	2 Days	4 Days	6 Days	8 Days	10 Days	
300	1.7 ± 1.6	36.7 ± 8.6	70 ± 15	85 ± 7.5	90 ± 5.8	100
250	5 ± 2.9	45 ± 8.7	65 ± 14.4	80 ± 11.5	85 ± 8.7	100
200	5 ± 2.9	37.5 ± 1.4	72.5 ± 1.4	77.5 ± 1.4	82.5 ± 1.4	100
150	0 ± 0	23.3 ± 6.3	55 ± 13.3	66.7 ± 13	73.3 ± 11.5	33
100	5 ± 5	23.3 ± 6.6	55 ± 12.5	65 ± 12.5	70 ± 10.4	33

**Table 3 biology-10-00233-t003:** Assessing the effect of CuO-NPs on wheat plant growth parameters including root length (cm), shoot length (cm), leaves number, root fresh weight (g), root dry weight (g), shoot fresh weight (g), and shoot dry weight (g).

Treatments/ ppm	Morphological Parameters						
	Root Length (cm)	Shoot Length (cm)	Leaves Number	Root Fresh Weight (g)	Root Dry Weight (g)	Shoot Fresh Weight (g)	Shoot Dry Weight (g)
**Control**	8.78 ± 0.21 ^b^	47.38 ± 0.82 ^b^	5.2 ± 0.2 ^b^	0.63 ± 0.04 ^b^	0.18 ± 0.01 ^c^	4.15 ± 0.04 ^b^	0.98 ± 0.03 ^c^
**Foliar CuO-NPs 50**	13.16 ± 0.57 ^a^	56.76 ± 0.68 ^a^	6.6 ± 0.4 ^a^	1.09 ± 0.07 ^a^	0.33 ± 0.02 ^a^	6.45 ± 0.05 ^a^	1.42 ± 0.04 ^a^
**Foliar CuO-NPs 100**	12.94 ± 0.58 ^a^	53.76 ± 1.97 ^a^	5.8 ± 0.2 ^ab^	1.04 ± 0.09 ^a^	0.29 ± 0.03 ^a^	5.82 ± 0.03 ^ab^	1.4 ± 0.05 ^a^
**Soaking CuO-NPs 50**	10.6 ± 0.48 ^b^	56.96 ± 0.9 ^a^	6.6 ± 0.4 ^a^	0.92 ± 0.12 ^ab^	0.28 ± 0.02 ^ab^	6.13 ± 0.04 ^a^	1.39 ± 0.04 ^ab^
**Soaking CuO-NPs 100**	9.82 ± 0.63 ^b^	54.92 ± 1.02 ^a^	6.4 ± 0.25 ^ab^	0.69 ± 0.05 ^b^	0.24 ± 0.02 ^bc^	6.01 ± 0.05 ^a^	1.24 ± 0.03 ^b^

Different letters in the same column indicate that mean values are significantly different (*p* ≤ 0.05) by Tukey’s test, means ±standard error (SE) (*n* = 5).

## Data Availability

The data presented in this study are available on request from the corresponding author.

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
