# Peer review of "Efficacy Assessment of Biosynthesized Copper Oxide Nanoparticles (CuO-NPs) on Stored Grain Insects and Their Impacts on Morphological and Physiological Traits of Wheat (Triticum aestivum L.) Plant"

_biology, 2021, doi:10.3390/biology10030233_

Round 1

Reviewer 1 Report

The manuscript entitled “Efficacy assessment of biosynthesized copper oxide nanoparticles (CuO-NPs) on stored grain insects and their impacts on morphological and physiological traits of wheat (Triticum aestivum L.) plant is interesting work as exploitation of microorganisms for the fabrication of nanoparticles (NPs) has garnered considerable research interest. However, numerous publications can be found in the literature on the biosynthesis of metal-based NPs using fungi. The authors missed several studies to discuss the finding with the risk associated with CuO-NPs such as doi.org/10.1007/s10653-020-00681-5. Keen attention should be paid to typos such as the very first line, it was written “cooper” etc, correct use of abbreviation (some places authors used NPs & nanoparticles, similar presentation of values and units are required throughout MS). There are many basic sentences in the “Introduction” which makes the weak presentation. For example, in the first paragraph, what authors want to state? It is expected to include a current overview of NPs production, application, and uses, then the work-related to CuO-NPs. Information on the wheat crop is fundamental and can be concise by addressing why they selected wheat? Line 83-84 needs citations. In which stage, authors collected samples for Morphological and Physiological measurements? And why selected such stage of wheat growth for analysis, pls mention in Materials and Methods section. Line 231-236 should be added in the Introduction section. Lines 92-94, 150-151, 349-351 are repeated, each section has such kind of repetition. In tables, the SD values are higher than the 10%. The authors did not mention "Statistical Analysis" detail in the Materials and Methods section

Author Response

Thank you very much for reviewing our manuscript, and for your valuable comments. Please see the attachment.

Reviewer 2 Report

Work done by Badawy et al was very nicely done.  It is very interesting work.  I especially like the work done on how the wheat grew in response to the nanoparticles.  I only had a few comments.  

Please spell out UV-Visible.  It is UV-Vis in many places.  Line 66 the sentence "wheat can grow..." seems to be incomplete.  If you filter the nanoparticles by size does the broadness go away in the UV-Visible spectrum? Can you comment more about the presence of Cu(I) and Cu(II)?  Are you making a Cu dimer nanoparticle or is it a mixture of different species of nanoparticles? Can you use DLS (Dynamic Light Scattering) to confirm the size range on the XRD?

Author Response

Thank you very much for reviewing our manuscript, and for your valuable comments. please see the attachment.

Reviewer 3 Report

The manuscript by Badawy et al. on CuO-NPs production by metabolites of A. niger funghi, and their spectroscopic and imaging characterization, as well as analysis of the activity as a biological against two wheat grain insects, and the effect on wheat plant growth and physiology has a logic structure, and the conclusions drawn are generally confirmed by the data presented. 

The manuscript needs English language check, particularly Summary, Abstract, and Introduction parts.

There are several points that need clarification:

  1. Material & Methods:
    1. Please provide more imaging parameters for TEM and SEM-EDX analysis, e.g. voltage, counts, spot or area analysis, tilt angle. Give the correct device information for SEM, as the description does not match with image data zone information. In addition, please provide more information on FT-IR spectroscopy parameters, e.g. no. of scans, reflectance or transmittance mode, baseline correction etc.
    2. Insecticidal bioassay: check the weight unit. It should be most reasonable x mg CuO-NPs mixed with 100 g of wheat grains.
  2. Results & Discussion:
    1. Please include a note on the different results from XRD (3 to 14 nm) and TEM (14 - 47 nm) for NP size and short discussion.
    2. SEM image shows considerably larger NPs or aggregates, please clarify. The image was made with BSE detector, please include short description ofwhat is seen (e.g. light areas: NP, dark areas: ?)
    3. FT-IR analysis: the conclusion "shifting in functional groups… " (l. 316-17) can not be drawn from data presented, and the sentence should be revised. In addition, no clear assignment of the wavelenght numbers to stretching vibrations of biological functional groups that would reduce and stabilize the CuO-NPS was made (e.g. naming relevant secondary metabolites). A spectrum of the metal precursor would be helpful to see changes after NP formation.
    4. Insecticidal bioassay: negative control is missing. If x mg per 100g wheat grain is meant, this results in 10 - 30 mg/kg or ppm. Why did the authors not studies the same range for the wheat plant growth assay, but instead higher concentrations of 50 and 100 mg/L or ppm?
    5. Plant growth assay: discussion is a bit confusing and needs to be refined. In the study presented, 50 ppm was low and 100 ppm was high concentration; but the cited literature described beneficial effect for lower values of 10 mg/L (=10 ppm) or up to 20 ppm only (no. 73 & 74). On the other hand, in l. 417 "lower CuO-NPs concentrations 100 mg/L" were beneficial in ref. 79, but presented data showed highest effect at 50 ppm. Moreover, please include a short discussion why foliar uptake of 50 ppm CuO-NPs is most effective compared to 50 ppm grain soaking.     

Author Response

(The authors gave the same response as above.)

Round 2

Reviewer 1 Report

The authors responded all comments. The manuscript can be accepted. 

Author Response

Reviewer comment #: The authors responded all comments. The manuscript can be accepted. 

Author response #: Thank you very much for your comment and agreement.

Reviewer 2 Report

I think this paper should be accepted.  Thank you for taking my comments under consideration

Author Response

Reviewer comment #: I think this paper should be accepted.  Thank you for taking my comments under consideration.

Author response #: Thank you for your comment and your agreement.

Reviewer 3 Report

The manuscript greatly improved over the V1. There are only few things to correct, sorry for insisting:

l. 32f Rewrite: FT-IR spectra identified functional groups of metabolites that could act as reducing, capping, and stabilizing agents to the CuO-NPs.

l. 139ff: The SEM device was now changed to Philips XL30-ESEM, but the data zone displays FEI Quanta 250FEG – which is more then 10 years newer on the market! It is obvious that the SEM image in Fig. 2B was NOT made with a XL30, so please provide the correct device! Additionally, it was not asked to give the SEM device's general specs, but to note the imaging parameters used, as they were added for TEM. According to the data zone of fig. 2B, at least 20 kV and BSE detector was used.

Please provide the EDX company (Oxford, Thermo Scientific, or?), and the analysis parameters (kV, WD, spot or area analysis, time or number of counts to collect spectra) – not the general specs of the device.

l. 304ff Should be corrected like this: In this study, the CuO-NPs size obtained by TEM analysis is bigger than those obtained by XRD. Each particle contains an amorphous and a crystal domain, and while TEM analysis results in the sum of these two domains as particle size, XRD only gives results for the crystal domain. Moreover, XRD measures the core of coated NPs, but not the surface coating, while TEM analysis measures overall NPs size [51].”

l. 520f: Should be corrected like this: …, we can conclude that 50 ppm of CuO-NPs exhibits more positive effects on… Foliar spraying appears to be more effective than soaking, which can be attributed to the spraying method ensuring a more uniform spreading over the crop canopy, in addition to a more immediate response from the crops [104].

Regards

Author Response

Reviewer comment #: l. 32f Rewrite: FT-IR spectra identified functional groups of metabolites that could act as reducing, capping, and stabilizing agents to the CuO-NPs.

Author response #: Thank you for your comment. Done

Reviewer comment #: l. 139ff: The SEM device was now changed to Philips XL30-ESEM, but the data zone displays FEI Quanta 250FEG – which is more than 10 years newer on the market! It is obvious that the SEM image in Fig. 2B was NOT made with an XL30, so please provide the correct device! Additionally, it was not asked to give the SEM device's general specs, but to note the imaging parameters used, as they were added for TEM. According to the data zone of fig. 2B, at least 20 kV, and a BSE detector was used.

Please provide the EDX company (Oxford, Thermo Scientific, or?), and the analysis parameters (kV, WD, spot or area analysis, time or number of counts to collect spectra) – not the general specs of the device.

Author response #: Thank you very much for your correction. We add the correct name of SEM and available data were added as follows: “The quantitative elemental composition of the fungal-mediated CuO-NPs was explored using Quanta 250FEG (Thermo Fisher Scientific, USA), (Accelerating voltage: 20.00 kV; magnification mode:3000x; with detector BSED). The Quanta FEG is integrated with an energy dispersive spectroscopy (EDX) detector (FEI company subsidiary of Thermo Fischer Scientific, Inc., Backscattered electron (BSE) detector; high electron beam at 2.5 nm at 30 KV)).”

Reviewer comment #: l. 304ff Should be corrected like this: In this study, the CuO-NPs size obtained by TEM analysis is bigger than those obtained by XRD. Each particle contains an amorphous and a crystal domain, and while TEM analysis results in the sum of these two domains as particle size, XRD only gives results for the crystal domain. Moreover, XRD measures the core of coated NPs, but not the surface coating, while TEM analysis measures overall NPs size [51].”

Author response #: Thank you very much for your correction. It is done.

Reviewer comment #: l. 520f: Should be corrected like this: …, we can conclude that 50 ppm of CuO-NPs exhibits more positive effects on… Foliar spraying appears to be more effective than soaking, which can be attributed to the spraying method ensuring a more uniform spreading over the crop canopy, in addition to a more immediate response from the crops [104].

Author response #: Thank you very much for your correction. It is correct as the reviewer recommends.

Finally, we hope the response meets the reviewer approval.